# Multiplication-Free Transformer Training via Piecewise Affine Operations

**Atli Kosson**     **Martin Jaggi**
EPFL, Switzerland
`firstname.lastname@epfl.ch`

## Abstract

Multiplications are responsible for most of the computational cost involved in neural network training and inference. Recent research has thus looked for ways to reduce the cost associated with them. Inspired by Mogami (2020), we replace multiplication with a cheap piecewise affine approximation that is achieved by adding the bit representation of the floating point numbers together as integers. We show that transformers can be trained with the resulting modified matrix multiplications on both vision and language tasks with little to no performance impact, and without changes to the training hyperparameters. We further replace all non-linearities in the networks making them fully and jointly piecewise affine in both inputs and weights. Finally, we show that we can eliminate all multiplications in the entire training process, including operations in the forward pass, backward pass and optimizer update, demonstrating the first successful training of modern neural network architectures in a fully multiplication-free fashion.

## 1   Introduction

The computational cost of training state-of-the-art neural networks has risen rapidly in recent years [1]. Aside from the increased price of training, this also requires more energy which often comes with an environmental impact. Large language models, such as foundation models [2], are particularly expensive to train. These models are typically based on the transformer architecture [34].

Neural network training consists largely of matrix multiplications that generally account for the vast majority of the computational cost for standard architectures such as transformers. These matrix multiplications are performed by multiplying and accumulating the scalar elements of the matrices which are generally encoded as floating point values during training. As floating point multiplication is significantly more complicated than addition, it requires more

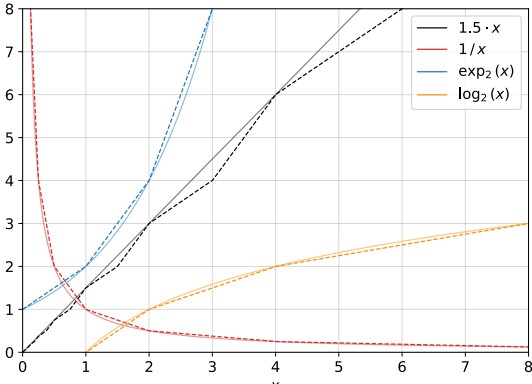

Figure 1: Elementary neural network operations (solid lines) along with cheaper piecewise affine alternatives (dashed lines). The piecewise affine functions periodically match the originals and the derivative of each segment is an exact power of 2.

logic gates and energy to perform in hardware. Horowitz (2014) estimates that for float32 operands, multiplication requires $4\times$ more energy than addition. Approaches for computationally efficient training thus often focus on reducing the cost of the multiplications. One such method is mixed precision training [23], where the multiplications are performed in a lower precision format such as float16 or bfloat16 [15] while the accumulation is still performed in the standard float32. Mod-

37th Conference on Neural Information Processing Systems (NeurIPS 2023).

ern hardware accelerators for deep learning, such as TPUs [14] and tensor core GPUs [26], have specialized hardware support for matrix multiplication in mixed precision.

A very different approach is to modify the neural network architecture itself, such as replacing multiplications with computationally cheaper but fundamentally different operations. This is the idea behind AdderNet [4] that uses the absolute difference of two numbers instead of their product in matrix multiplications. AdderNets involve few multiplications during inference but require special tricks during backpropagation that involve multiplications and/or exponentiation to achieve optimal performance. Another alternative is tropical algebra that replaces the products with additions and the accumulation with a maximum, but has not shown to give performance competitive with standard neural networks [22, 9]. For both of these approaches, the main hardware advantage is replacing multiplications with additions that tend to be much cheaper in hardware.

The core idea we pursue in this paper falls in-between the two aforementioned approaches. We replace multiplications with operations that roughly approximate them but have a cost as cheap as addition. Since the operations are not fundamentally different, they should serve as a drop-in replacement for multiplications—similar to mixed precision—without requiring architectural changes, while potentially being significantly cheaper. One such approximation is Mitchell's logarithmic multiplication [24] which approximates the product of $A > 0$ and $B > 0$ as:

$$AB \approx \mathrm{paexp}_2(\mathrm{palog}_2(A) + \mathrm{palog}_2(B)) \tag{1}$$

where $\mathrm{paexp}_2$ and $\mathrm{palog}_2$ are cheap approximations of $\exp_2$ and $\log_2$ shown in Figure 1 along with an example output. All the approximations are *piecewise affine* (sometimes called piecewise linear) in their input arguments. The core of this operation can be performed efficiently in hardware using int additions which are around $9\times$ cheaper than float addition for 32-bit operands [12]. We roughly estimate the cost of piecewise affine multiplication to be around $30\times$ less than that of standard float32 multiplication in terms of the area and energy used by the operation itself (Appendix B)[1]. Approaches based on logarithmic multiplication have also been explored for neural network inference and found to enable around $5\times$ energy saving compared to bfloat16 multiply-and-accumulate operations [16].

Mogami [25] showed that Equation (1) can be performed by directly adding the floating point number representations together as integers, with some extra handling of the exponent accounting for the exponent bias, underflows and overflows. They further show that this trick can be used to train convolutional neural networks (in particular ResNet50 [11]) with a minor or no degradation compared to standard multiplications. In this work we further explore the application of piecewise affine approximations to neural network training, summarized below:

- We show that transformers can be trained with piecewise affine matrix multiplications on both vision and language data with little to no performance impact. We compare this to AdderNet based transformers [30] demonstrating better accuracy while replacing more multiplications.

- We define additional piecewise affine functions for all other operations in a neural network and demonstrate fully multiplication-free training, with only a minor performance impact. No multiplications are used in any part of training, including the forward pass, backpropagation and optimizer computation. To the best of our knowledge, this is the first time a neural network has been trained entirely without standard multiplications.

- We show that the resulting neural networks are fully piecewise affine (sometimes called piecewise linear) both in their input and in their weights, a property that traditional transformers are very far from satisfying.

- We publicly release our code[2], including custom kernels, in the hopes of aiding further research into multiplication-free neural networks.

---

[1]Note that special hardware support is needed to realize these gains, see discussion in Section 5

[2]Code available at `https://github.com/epfml/piecewise-affine-multiplication`

## 2 Methods

### 2.1 Floating Point Numbers

Since the exact representation of floating point numbers is important to our method we will give a brief overview here. A floating point number is represented as:

$$(-1)^S \cdot 2^E \cdot (1 + M) \tag{2}$$

where $S \in \{0, 1\}$ is a sign bit, $E$ is an integer encoding the exponent, and $0 \leq M < 1$ is the mantissa fraction. Note that this is analogous to the normalized scientific notation for decimal numbers (for example $+1.23 \cdot 10^{-2}$).

The encoding used for $E$ and $M$ varies between floating point formats. The commonly used IEEE 754 float32 uses the following bit representation:

$$[S, \underbrace{e_0, \ldots, e_7}_{=: \bar{E}}, \underbrace{m_0, \ldots, m_{22}}_{=: \bar{M}}] \tag{3}$$

where the exponent is encoded as an unsigned integer $\bar{E}$ with a bias of 127 i.e. $E = \bar{E} - 127 = e_0 e_1 e_2 e_3 e_4 e_5 e_6 e_7 - 127$ and the mantissa as an unsigned integer $\bar{M}$ that is divided by $2^{23}$ to get $M = \frac{\bar{M}}{2^{23}} = \frac{m_0 \ldots m_{22}}{2^{23}}$.

Floating point numbers can also encode special values that do not fit within the representation given in Equation (2). In float32 $\bar{E} = 255$ is used to represent NaN (not a number) or infinity depending on the value of $\bar{M}$. For $\bar{E} = 0$ the float is interpreted as a denormal number $(-1)^S \cdot 2^E \cdot M$ which also gives the representation of 0, which is $\bar{E} = 0$ and $\bar{M} = 0$. Some floating point formats, e.g. bfloat16 do not allow denormal numbers other than 0, since those can be more expensive to handle in hardware.

### 2.2 Piecewise Affine Multiplication

We define the *piecewise affine multiplication* (or *PAM* for brevity) of two floating point numbers:

$$A = (-1)^{S_A} \cdot 2^{E_A} \cdot (1 + M_A), \qquad B = (-1)^{S_B} \cdot 2^{E_B} \cdot (1 + M_B) \tag{4}$$

as $A \,\hat{}\, B$:

$$A \,\hat{}\, B := (-1)^{S_{A \hat{} B}} \cdot 2^{E_{A \hat{} B}} \cdot (1 + M_{A \hat{} B}) \tag{5}$$

$$S_{A \hat{} B} := S_A + S_B \tag{6}$$

$$E_{A \hat{} B} := E_A + E_B + \mathbf{1}\{M_A + M_B \geq 1\} \tag{7}$$

$$M_{A \hat{} B} := M_A + M_B - \mathbf{1}\{M_A + M_B \geq 1\} \tag{8}$$

where $\mathbf{1}\{a\}$ is one when $a$ holds and zero otherwise.

We observe that $A \,\hat{}\, B$ is roughly achieved by adding the signs, exponents and mantissas of $A$ and $B$. In fact, this becomes precise if we allow $M_{A \hat{} B}$ to overflow into $E_{A \hat{} B}$. This happens if we add the float32 bit representations of $A$ and $B$ together as int32.

A technicality is that the exponent requires special handling, as the resulting $\bar{E}$ is only offset by $-127$ but should be offset by twice that amount requiring us to subtract 127 from the exponent of the sum. We also need to make sure the resulting $\bar{E}$ has not overflowed or underflowed, instead clamping it to the min and max values. Possible denormal values (mantissa starting with zeros) can be flushed to zero for simplicity, as done in e.g. bfloat16. Finally we check that the incoming $A$ and $B$ do not represent special values such as NaN or infinity and handle those cases explicitly.

For float32, PAM can thus be simply implemented with two int additions and a couple of checks for the exponent. For current hardware accelerators such as GPUs, this is expensive and requires multiple instructions. However the logic is much simpler than for standard multiplication, so new hardware with native support could perform this operation cheaply. With a custom floating point format without an exponent bias, we could further get rid of one of the int additions as well. This could also be done for configurable hardware such as FPGAs. In Appendix B we roughly estimate the hardware cost of PAM, estimating the savings to be around 5-30x over standard multiplication in float16 and float32.

## 2.3 Other Piecewise Affine Operations

As mentioned in the introduction, piecewise affine multiplication can be expressed in terms of $\mathrm{paexp}_2$ and $\mathrm{palog}_2$ which we can write as follows for $A$, $B$ from Equation (4):

$$\mathrm{paexp}_2(A) = 2^{\lfloor A \rceil} \cdot (1 + A - \lfloor A \rceil) \tag{9}$$

$$\mathrm{palog}_2(A) = E_A + M_A, \text{ for } A > 0 \tag{10}$$

where $\lfloor A \rceil$ denotes rounding down to the nearest integer. For positive inputs Equation 1 then becomes:

$$AB \approx \mathrm{paexp}_2(\mathrm{palog}_2(A) + \mathrm{palog}_2(B)) \tag{11}$$

$$= 2^{\lfloor E_A + E_B + M_A + M_B \rceil} \cdot \left(1 + E_A + E_B + M_A + M_B - \lfloor E_A + E_B + M_A + M_B \rceil\right) \tag{12}$$

$$= 2^{E_A + E_B + \mathbf{1}\{M_A + M_B \geq 1\}} \cdot \left(1 + M_A + M_B - \mathbf{1}\{M_A + M_B \geq 1\}\right) \tag{13}$$

Which is equivalent to $A \,\hat{\ast}\, B$, aside from the computation of the sign.

We define piecewise affine division as the inverse of $A \,\hat{\ast}\, B$:

$$A \,\hat{\div}\, B := (-1)^{S_{A \hat{\div} B}} \cdot 2^{E_{A \hat{\div} B}} \cdot (1 + M_{A \hat{\div} B}) \tag{14}$$

$$S_{A \hat{\div} B} := S_A - S_B \tag{15}$$

$$E_{A \hat{\div} B} := E_A - E_B - \mathbf{1}\{M_A - M_B \leq 0\} \tag{16}$$

$$M_{A \hat{\div} B} := M_A - M_B + \mathbf{1}\{M_A - M_B \leq 1\} \tag{17}$$

which corresponds to $\mathrm{paexp}_2(\mathrm{palog}_2(A) - \mathrm{palog}_2(B))$ for positive numbers. $A \,\hat{\div}\, B$ can also be efficiently performed in hardware float32 using int32 subtraction where a bias offset of 127 needs to be added to the result.

With $\mathrm{paexp}_2$ and $\mathrm{palog}_2$ we can also define general piecewise affine functions for powers, including the natural logarithm and exponential as well as square roots.

$$\mathrm{paexp}(A) := \mathrm{paexp}_2(\log_2(e) \,\hat{\ast}\, A) \tag{18}$$

$$\mathrm{palog}(A) := \mathrm{palog}_2(A) \,\hat{\div}\, \log_2(e) \tag{19}$$

$$\mathrm{pasqrt}(A) := \mathrm{paexp}_2(\mathrm{palog}_2(A) \,\hat{\div}\, 2) \tag{20}$$

With these functions we can implement various neural networks operations such as normalization layers, softmax, cross-entropy and other losses as well as optimizer updates.

## 2.4 Piecewise Affine Neural Networks

Modern artificial neural networks are compositions of the operations discussed in the previous subsections. We thus replace each traditional operation in a network with its piecewise affine approximation. Unlike traditional networks, the resulting network will now be a piecewise affine function, both in its weights and also in its input values. This follows directly since the composition (and sum) of two piecewise affine functions (with one or many arguments) remains piecewise affine. We iterate the composition through all layers and the loss computation.

This global property of being piecewise affine (and thus having piecewise constant gradients) is remarkable as it is does not hold for traditional e.g. ReLU networks, due to the composition with the most common losses, and also is far from true for traditional transformers. Despite this drastic change in the gradient structure and loss landscape, we will show that commonly used optimizers (and their multiplication-free counterparts) still enable successful training out of the box without additional hyperparameter tuning.

## 2.5 Derivatives

Following Mogami [25], we consider two ways to compute the derivatives of the piecewise affine functions. The first option is to compute the true derivatives of those functions, which are piecewise constant and discontinuous. The second option is to implement the analytical derivative of the original function we are approximating, for example standard multiplication, with the piecewise affine functions. We will refer to the first option as the *exact derivative* and the second as the

Table 1: Derivatives for Piecewise Affine Operations for A, B defined in Equation 4.

| OPERATION | EXACT DERIVATIVE | APPROXIMATE DERIVATIVE |
|---|---|---|
| $Y = A \mathbin{\hat{*}} B$ | $\delta_A = 2^{E_B + \mathbf{1}\{M_A + M_B \geq 1\}} \delta_Y$ | $\delta_A = B \mathbin{\hat{*}} \delta_Y$ |
| $Y = A \mathbin{\hat{\div}} B$ | $\delta_A = 2^{-E_B - \mathbf{1}\{M_A - M_B \leq 0\}} \delta_Y$ | $\delta_A = \delta_Y \mathbin{\hat{\div}} B$ |
| $Y = A \mathbin{\hat{\div}} B$ | $\delta_B = -(A \mathbin{\hat{*}} \delta_Y) \mathbin{\hat{\div}} (B \mathbin{\hat{*}} B)$ | $\delta_B = -(A \mathbin{\hat{*}} \delta_Y) \mathbin{\hat{\div}} (B \mathbin{\hat{*}} B)$ |
| $Y = \mathrm{paexp}_2(A)$ | $\delta_A = 2^{\lfloor A \rfloor} \delta_Y$ | $\delta_A = 2^A \mathbin{\hat{*}} \ln(2) \mathbin{\hat{*}} \delta_Y$ |
| $Y = \mathrm{palog}_2(A)$ | $\delta_A = 2^{-E_A} \delta_Y$ | $\delta_A = \delta_Y \mathbin{\hat{\div}} (A \mathbin{\hat{*}} \ln(2))$ |

*approximate derivative*. Table 1 lists both kinds of derivatives for each operation described previously. We use delta notation for backpropagation, i.e. $\delta_A = \frac{\partial L}{\partial A}$ and $\delta_Y = \frac{\partial L}{\partial Y}$ for some unspecified scalar function $L$ that only depends on $A$ through the output $Y$.

We note that even though the exact derivative sometimes contains a multiplication, one of the numbers is an exact power of two which means that the multiplication can be performed exactly via PAM, see Section 2.7. Both types of derivatives can therefore be computed in a multiplication-free manner, only using PAM and related operations. This also extends to derived operations such as paexp, palog, pasqrt, softmax and normalization. We can backpropagate through the derived functions via the the computational graph that defines them, using either the exact or approximate derivatives given above. By extension, we can perform the entire forward and backward pass through a neural network consisting of these primitives without any traditional multiplication operations.

## 2.6 Optimizers

Networks with piecewise-affine components can be trained using standard gradient based optimization using the derivatives given in the previous section. However, the implementation of the optimization algorithm itself might involve additional multiplications (such as with varying learning rates, adaptive learning rates or for computing other optimizer statistics). We thus also experiment with replacing all optimizer multiplications and divisions with their piecewise affine analogs, and quantify the impact of this change in our experimental results.

## 2.7 Approximation Error

Figure 2 compares PAM and standard multiplication. It also shows the relative error of $x_1 \mathbin{\hat{*}} x_2$ compared to $x_1 x_2$, which depends on the exact mantissas of the inputs. PAM gives the exact result when the mantissa of either input is zero, i.e. when it is a power of 2. The maximum relative error of $\frac{(1 + 0.5 + 0.5) - 1.5^2}{1.5^2} = -1/9 = -11.1\%$ is obtained when $M = 0.5$ for both numbers. When either input is a fixed, e.g. when multiplying a variable by a constant, the maximum error can be lower.

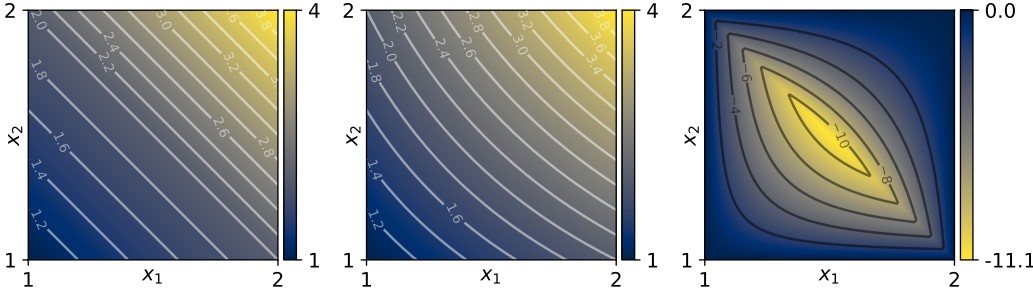

Figure 2: Comparison of the piecewise affine multiplication and standard multiplication of two numbers $x_1, x_2$ in the range $[1, 2]$. These patterns repeat per octave. **Left:** Piecewise affine multiplication. **Middle:** Standard multiplication. **Right:** Relative error $\frac{x_1 \mathbin{\hat{*}} x_2 - x_1 x_2}{x_1 x_2}$ measured in percents.

The relative approximation error of PAM can be reduced by an additional PAM operation. For an appropriately choosen value $\alpha$, the expression $x_1 \,\hat{\cdot}\, x_2 \,\hat{\cdot}\, \alpha$ will better approximate $x_1 x_2$ on average and/or the worst case. If either $x_1$ or $x_2$ is a constant that is known ahead of time, e.g. when computing $\mathrm{paexp}$ and $\mathrm{palog}$, this can be taken into account when choosing $\alpha$ for a specific operation. For PAM matrix multiplications, scaling the output instead of the inner scalar products may be sufficient to eliminate bias in the output magnitude. In general this sort of approximation will be more expensive than a single PAM operation and we leave it to future work to explore where and when this additional error compensation is beneficial.

The exact and approximate derivatives have different error characteristics when viewed as approximations of the derivative for a standard multiplication. The exact derivative is unbiased on average, since PAM is continuous and matches multiplication at certain points. However, the exact derivatives are discontinous and have a high variance in the sense that for a specific input the error is often large. On the other hand, the approximate derivative usually has a lower error at a given point (i.e. low variance) but may be biased on average. These characteristics of the two derivative types also hold for the other functions. Figures 3 and 4 in the appendix show the exact and approximate derivatives for several functions.

## 3   Experiments & Results

### 3.1   Experimental Setup

We use PyTorch [28] for our experiments and run them using either Nvidia A100 (40GB) or V100 (32GB) GPUs. Elementwise piecewise affine multiplication, division, $\mathrm{palog}_2$ and $\mathrm{paexp}_2$, as well as PAM matrix multiplications are performed using custom CUDA kernels. For verification, we also run alternative native torch implementations of these operations that run significantly slower. Other functions such as normalization and softmax are implemented by composing these operations. The training setup (e.g. hyperparameters) for both the standard and piecewise affine networks is identical. All hyperparameters are obtained from other works and have not been tuned specifically for piecewise affine training.

In our experiments we consider two training tasks, one on text data and one in computer vision. The first one is German to English translation on the IWSLT14 DE-EN dataset [3] consisting of roughly 160K sentence pairs from translated TED talks. The model is a small transformer with 6 encoder and decoder layers each, 4 attention heads an embedding dimension of 512, a feed forward hidden dimension of 1024 and ReLU activations. The training code for this task is based on the FairSeq [27] codebase. Our baseline setup trains for 20 epochs, using a cosine decay schedule with 4000 warmup steps and a peak learning rate $5 \cdot 10^{-4}$ with a maximum batch size of 4096 tokens. We use AdamW [21, 17] for optimization with $\beta_1 = 0.9$, $\beta_2 = 0.98$ and weight decay of $10^{-4}$. During training we apply a dropout with drop probability 0.3 and use cross entropy with a label smoothing of 0.1. For evaluation we use a beam search with 5 beams and report the BLEU score on the test set.

The second task is training a DeiT-Tiny [32] Vision Transformer [5, 7] without distillation. This model has 5M parameters, 12 layers with 3 heads and an embedding dimension of 192. We train on either CIFAR10 [19] or the ImageNet-1k [6] dataset. CIFAR10 consists of 50K training and 10K test images of size $32 \times 32$ corresponding to 10 classes. When training on CIFAR10 we upscale the images to $224 \times 224$ and use the same data augmentation procedure described in DeiT [32] for ImageNet. ImageNet-1k [6] contains 1.28M training images and 50K validation images in 1000 classes. The images have different sizes but are processed at a resolution of $224 \times 224$ for training. These experiments are based on the PyTorch Image Models project [36]. Optimization is performed using AdamW with $\beta_1 = 0.9$, $\beta_2 = 0.999$ and weight decay of 0.05. Our base learning rate of $5 \cdot 10^{-4}$ which is scaled linearly by the total batch size (1152 for ImageNet, 1024 for CIFAR10) divided by 512. We use a cosine decay learning rate schedule with a warmup of 5 epochs and train for a total of 300 epochs on ImageNet or 600 epochs on CIFAR10.

### 3.2   Replacing Matrix Multiplications

As mentioned before, matrix multiplications account for the majority of the computational cost of training neural networks and are therefore provide the largest opportunity for cost reduction. We experiment with replacing all matrix multiplications on both the forward and the backward passes with matrix multiplications where scalar multiplications are replaced by piecewise affine

Table 2: Top-1 Test/Validation Accuracy for DeiT-Tiny Training. The brackets denote the change from the baseline. *Shu et al. 2021 reports a different baseline accuracy of $92.6\%$ on CIFAR10.

| DATASET | BASELINE | PA-MATMUL | ADDER |
|---|---|---|---|
| CIFAR10 | 94.8% | 95.0[+0.2]% | 92.4[−0.2]%* |
| IMAGENET | 72.2% | 72.2[+0.0]% | 70.5[−1.7]% |

multiplications. This includes all linear layers, the batched matrix multiplication used in the attention and the convolutional layer used for the patch embedding in the vision transformer. Here we use the same type of matrix multiplications on the backwards pass which corresponds to the approximate derivatives described in Section 2.5.

Our results for DeiT-Tiny on CIFAR10 and ImageNet can be seen in Table 2. The PAM based matrix multiplication training matches the baseline accuracy for both datasets. We compare to Adder Attention [30] which also replaces multiplications by cheaper addition based operations. They do not replace the batched matrix multiplication and use standard matrix multiplication for the first and last transformer layer on ImageNet and also use multiplications to smooth the gradients during the backwards pass. Adder Attention transformers can roughly match their reported baseline accuracy for CIFAR10, but do suffer from a sizable degradation on ImageNet.

For IWSLT14 we obtain a baseline test BLEU score of $34.37$. Replacing all matrix multiplications on the forward and backwards passes with PAM based matrix multiplications gives a test BLEU score of $34.22$. Both scores are averaged over three runs. The observed difference of $0.15$ is slightly larger than the run-to-run variance we observed, but fairly small overall. We also note that we use standard hyperparameters originally obtained from FairSeq and tuning the hyperparameters may improve performance.

In Appendix C we show similar experiments for several non-transformer models including convolutional neural and mixer architectures on CIFAR-10. We find little to no degradation across all networks tested.

### 3.3 Replacing Other Operations

In Table 3 we measure the impact of using piecewise affine versions of different network components. We isolate the effect of each component and experiment with the two different derivative types described in Section 2. MATMUL refers to the linear layers and the batched matrix multiplications that we replaced in Section 3.2. Using exact derivatives for the matrix multiplications performs notably worse than the approximate ones. The same holds for piecewise linear layer normalization, with approximate derivatives we observe little to no degradation but using the exact ones cause a notable one. ATTENTION SOFTMAX refers to the softmax operation performed in the attention computation and excludes the softmax for the loss. Here we find that the using the exact derivatives cause a training instability when using the default hyperparameters whereas the with the approximate backwards pass the performance impact is negligible. The LOSS function, softmax cross-entropy with label smoothing, is the only operation where we find the approximate BWD pass to perform

Table 3: BLEU Test Scores on IWSLT14 DE-EN translation for modified Transformer-Small training where certain operations are replaced with piecewise affine approximations using either the exact or approximate backward pass. Cumulative experiments replace all previously listed operations using the better performing derivative version for each one. The brackets denote the change from the baseline BLEU score of $34.4 \pm 0.1$. Each reported score is the average±std over three runs with different seeds.

| PA OPERATION(S) | EXACT BWD | MIMIC BWD | CUMULATIVE |
|---|---|---|---|
| MATMUL | 33.4[−1.0] ± 0.1 | 34.2[−0.2] ± 0.2 | 34.2[−0.2] ± 0.2 |
| ATTENTION SOFTMAX | 3.0[−31.4] ± 0.9 | 34.3[−0.1] ± 0.7 | 34.2[−0.2] ± 0.2 |
| LAYER NORM | 33.7[−0.7] ± 0.2 | 34.4[−0.0] ± 0.1 | 34.4[−0.0] ± 0.1 |
| LOSS | 33.7[−0.7] ± 0.2 | 33.0[−1.4] ± 0.9 | 33.7[−0.7] ± 0.2 |
| OPTIMIZER | 34.2[−0.2] ± 0.1 | | 33.5[−0.9] ± 0.3 |

worse than the exact one. It is also the only network component where both types of derivatives have a notable performance loss.

Finally we experiment with using piecewise affine operations for the optimizer update. The AdamW optimizer uses multiplication, division and square root. Replacing all of them only causes a minor degradation of 0.2 BLEU compared to the standard version.

### 3.4 Fully Multiplication-Free Training

Here we combine all the piecewise affine replacements from the previous section one by one. The resulting BLEU scores are displayed in the last column of Table 3. We chose the derivative type that performed better for each operation which is the approximate one for everything except the loss. Each transformer block contains a gain that we replace with PAM when we replace the attention softmax.

Overall there is little to no performance impact from replacing the matrix multiplications, softmax and layer normalization. Replacing all of them together does not seem to cause any unforseen issues. Consistent with our previous observations, replacing the loss with a piecewise affine function causes a small BLUE score degradation.

The last entry in Table 3 covers every multiplication and non-linearity in the training process. This includes the forward pass, the backwards pass and the parameter updates by the optimizer. The resulting performance impact is only 0.9 BLEU points and hyperparameter tuning could potentially reduce it even further.

## 4 Related Work

As mentioned in the introduction, most related work falls into the two classes of approaches, depending if the original neural network architecture is preserved or not.

For the first type, fine-grained changes of operations are made to individual multiplications or groups thereof, aiming to closely approximate the original multiplications. Examples for this are mixed precision training [23], and more generally block-floating point schemes, where several weights are grouped together to share the same floating point exponent, enabling inference and training [18, 8]. Another class of related approaches is quantization, which is widely used for inference but introduces additional challenges for training [20]. All mentioned approaches have in common that some multiplications are still needed, thus might not be viable on multiplication-free hardware.

Approaches of the second type allow more drastic changes to the architecture, thus offering more promise to new types of multiplication-free operations. A prominent example are AdderNets [4, 30] or the related EF-operator [38] which rely on absolute differences instead of inner products, and show these can serve as replacement layers for convolutions or also parts of self-attention layers [30], though training can not be made fully multiplication-free while maintaining performance. Another type of fundamentally different architectures can arise by replacing classical arithmetic with tropical algebra, where all multiplications are replaced by additions and additions are replaced by maximum. While numerically very efficient, performance was not shown to be competitive with standard neural networks [22, 9].

To the best of our knowledge, the approach of Mogami [25]—which serves as the basis of our work here—is currently the only work which allows to preserve the neural network architecture while at the same time paving the way for training on fully multiplication-free hardware, if applied to all elements of the training process.

## 5 Discussion

The main goal of this work is to demonstrate the applicability of piecewise affine function approximations to end-to-end transformer training. In our experiments we showed that all operations during training can be replaced by such functions. We believe this is the first fully multiplication-free training of a transformer model and potentially any neural network. While PAM approximates real multiplication to a certain extent, it does not aim to do so as closely as possible, unlike floating point multiplication in finite precision. Instead PAM uses line segments with slopes that are powers of 2. By doing so it becomes significantly simpler, providing the same hardware benefit as replacing all multiplications with e.g. addition, like some other approaches to multiplication-free networks aim to

do. At the same time PAM does roughly approximate standard multiplications overall, allowing it to function as a plug-in replacement across various usecases. This includes the optimizer update and the weighted average in attention, which would be hard to replace with something like addition. PAM therefore seems to strike a good balance between cost and utility.

Our main inspiration is the work of Mogami [25] which applied piecewise affine approximations to convolutional networks, including convolutional layers, linear layers, batch normalization and exponents. Depending on the implementation, this may have been sufficient to make the forward and backwards pass fully multiplication-free but it is unfortunately unclear from the paper and no code is publicly available. We focus on transformers making both the forward and backwards passes multiplication-free and fully piecewise affine. We also extend this to the optimizer eliminating all multiplications in the training process, allowing it to run on entirely on hardware without multiplication support.

Although piecewise affine operations are logically simpler than their standard equivalents, special hardware is required to fully reap the theoretical advantages. Currently available hardware accelerators, such as GPUs and TPUs, have arithmetic units that are specialized for floating point multiplication but lack the comparatively simpler logic circuits that would enable fast piecewise affine multiplication. In practice this means that our present GPU implementation has to perform multiple instructions for every standard multiply-and-accumulate operation causing it to run significantly slower, even with custom kernels. Field-Programmable Gate Arrays (FPGAs) are one type of hardware that could immediately benefit from PAM. Since their logic circuits are fully configurable, they don't have the "sunken cost" of standard multiplication arithmetic and can instead spend it on PAM circuits.

As described in Section 2.4, replacing all multiplications and non-linearities in a neural network with piecewise affine operations results in a network that is fully jointly piecewise affine in both inputs and weights. This is an interesting theoretical property since it differs significantly from standard transformers.

Surprisingly, replacing all operations in the network itself doesn't seem to have a large impact on the performance in our setup. This differs from the observation by Mogami 2020 which found that linear layers did not work well with approximate derivatives. We observe a similar phenomenon for the loss function which could indicate that training is sensitive to slight errors early in backpropagation.

This project focused on standard neural networks without any modifications. However it would be interesting to explore modifying the network architecture to be even better suited to piecewise affine approximations. One potential way of doing this is to use $\log_2$ and $\exp_2$ throughout, for instance in the softmax and loss computation. These operations are much easier to approximate than their base-$e$ equivalents. Exploring the effects of error both from quantization as the mantissa is decreased and from magnitude bias is another future research direction.

## 6  Conclusion

In this work we explored the use of hardware-efficient piecewise affine operations for transformer training. We experiment on CIFAR10 [19] and ImageNet-1k [6] with DeiT-tiny [32] as well as on IWSLT14 German to English translation [3] using a small transformer network. We show that floating point multiplications within matrix multiplications can be replaced by cost-effective piecewise affine multiplications with minor or no performance degradation for all three datasets, and without additional hyperparameter tuning.

We further show that we can extend this to replace all multiplications, division and non-linear functions in the small transformer, the loss function and the optimizer with piecewise affine functions. The resulting performance impact is minor and could potentially be reduced further by hyperparameter tuning. This eliminates all multiplications in the entire training process, for fully multiplication-free training which we believe has not been demonstrated before. This also makes the network fully and jointly piecewise affine in both inputs and weights, a property that could be of theoretical interest.

We publicly release our code, including the kernels that made our experiments computationally viable. This could help further research in this area and into other multiplication-free methods.

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

# A Additional Figures

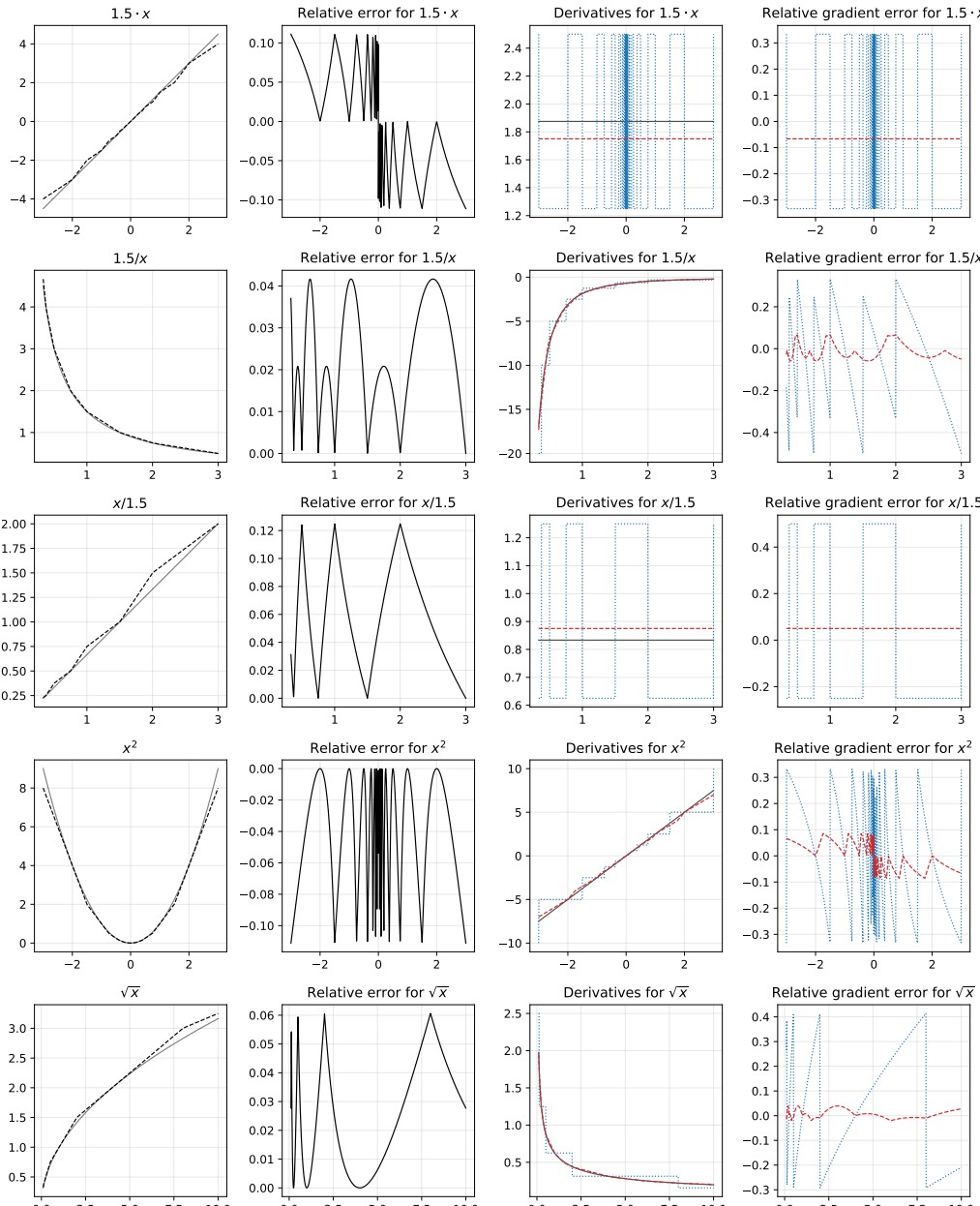

Figure 3: Piecewise affine multiplication, division, square and square root along with their standard equivalents. Each row corresponds to one function. **Left:** A function (solid line) with its piecewise affine alternative (dashed line). **Left middle:** The relative error $\frac{\hat{f}(x)-f(x)}{|f(x)|}$ when the piecewise affine function $\hat{f}$ is viewed as an approximation of the standard function $f$. **Right middle:** The derivatives of $f$ (solid black), the exact derivative for $\hat{f}$ (dotted blue) and the approximate derivative of $\hat{f}$ (dashed red). We assume $\delta_Y = 1.25$ in the equations in Section 2.5 which emphasizes the PAM operations used, compared to $\delta_Y = 1$ where PAM matches standard multiplication. **Right:** The relative error $\frac{\hat{f}'(x)-f'(x)}{|f'(x)|}$ for the exact derivative $\hat{f}'$ (dotted blue) and the approximate derivative $\hat{f}'$ (dashed red) when viewed as an approximation of $f'(x)$, the derivative of $f(x)$.

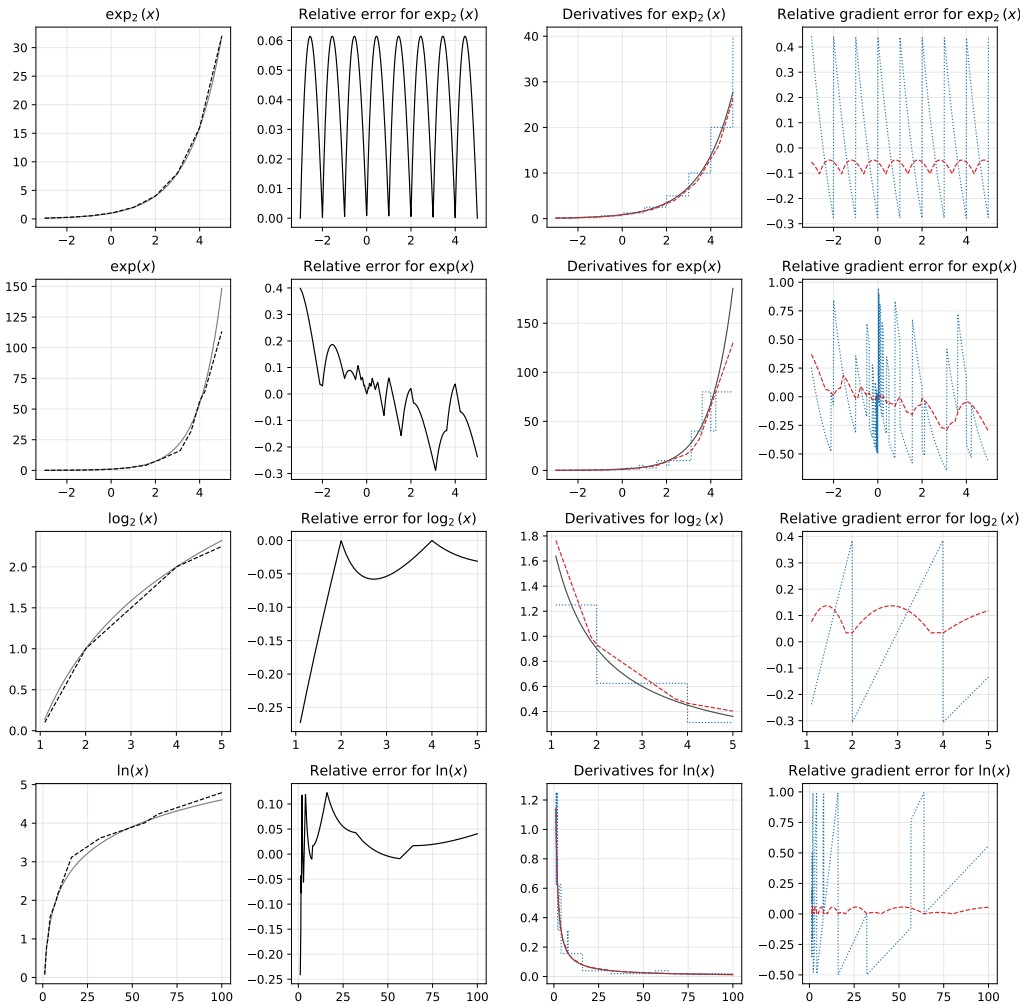

Figure 4: The equivalent of Figure 3 for exponentials and logarithms, showing piecewise affine functions with their standard counterparts. Piecewise affine exponentials and logarithms along with their standard equivalents. Each row corresponds to one function. **Left:** A function (solid line) with its piecewise affine alternative (dashed line). **Left middle:** The relative error $\frac{\hat{f}(x)-f(x)}{|f(x)|}$ when the piecewise affine function $\hat{f}$ is viewed as an approximation of the standard function $f$. **Right middle:** The derivatives of $f$ (solid black), the exact derivative for $\hat{f}$ (dotted blue) and the approximate derivative of $\hat{f}$ (dashed red). We assume $\delta_Y = 1.25$ in the equations in Section 2.5 which emphasizes the PAM operations used, compared to $\delta_Y = 1$ where PAM matches standard multiplication. **Right:** The relative error $\frac{\hat{f}'(x)-f'(x)}{|f'(x)|}$ for the exact derivative $\hat{f}'$ (dotted blue) and the approximate derivative $\hat{f}'$ (dashed red) when viewed as an approximation of $f'(x)$, the derivative of $f(x)$.

Table 4: Hardware costs of arithmetic operations in different precision from [12, 10].

| | ADDITION | | MULTIPLICATION | |
| FORMAT | ENERGY [pJ] | AREA [$\mu m^2$] | ENERGY [pJ] | AREA [$\mu m^2$] |
|---|---|---|---|---|
| INT32 | 0.1 | 137 | 3.1 | 3495 |
| INT16 | 0.05 | 67 | | |
| INT8 | 0.03 | 36 | 0.2 | 282 |
| FLOAT32 | 0.9 | 4184 | 3.7 | 7700 |
| FLOAT16 | 0.4 | 1360 | 1.1 | 1640 |

## B    Hardware Cost of Piecewise Affine Multiplication

It is important to note that our objective in this work is to investigate the machine learning characteristics of training with piecewise affine operations, rather than designing a hardware implementation. We demonstrate that such training is viable, with final model performance and convergence speed roughly identical to standard training. Without such algorithmic evidence to motivate their development, the development of efficient hardware implementations is unlikely. Although the details of the integrated circuit design should not affect e.g. convergence, they can drastically affect the cost of the operation in terms of energy or area.

Such hardware considerations include which aspects of standard floating point numbers are supported, e.g. denormal numbers, different types of NaNs and so on, as well as whether to use a custom encoding of the exponent for instance. An actual hardware implementation would also have to consider factors such as latency, as well as memory characteristics like bandwidth and read costs. The design could either focus on individual operands or vectors like tensor cores and systolic arrays. Operating directly on vectors or matrices can allow significant amortization of some of the costs involved.

Ignoring most of these considerations we can arrive at a very rough estimate of the costs. We assume that the cost of multiply-and-accumulate operations is the sum of the costs provided in Table 4 for multiplication and addition. A PAM operation can be performed with one full int32 addition and one int8 addition for the exponent which also needs to check for under / overflow. When implemented as a logic circuit, we estimate the cost of this could be comparable to two int32 additions. Using these numbers float32 PAM uses roughly $\frac{2 \cdot 0.1}{3.7} = 5.4\%$ of the energy and $\frac{2 \cdot 137}{7700} = 3.6\%$ of the area compared to float32 multiplication and $\frac{2 \cdot 0.1}{1.1} = 18\%$ of the energy and $\frac{2 \cdot 137}{1640} = 17\%$ of the area compared to float16 multiplication. PAM is likely to work with smaller numbers such as float16 which would reduce the cost further by a factor of around $2\times$. In Appendix D we find that mantissas as narrow as 4 bits may work well, significantly less than the 10 and 7 bits used in float16 and bfloat16.

PAM reduces the cost of multiplications by roughly an order of magnitude but does not affect accumulation. The average savings for multiply-and-accumulate (MAC) operations will therefore be lower. For float32-multiply float32-accumulate PAM could have a cost of roughly $\frac{2 \cdot 0.1 + 0.9}{3.7 + 0.9} = 24\%$ of the energy and $\frac{2 \cdot 137 + 4184}{7700 + 4184} = 38\%$ of the area. For standard mixed precision float16-multiply float32-accumulate the costs are $\frac{2 \cdot 0.1 + 0.9}{1.1 + 0.9} = 55\%$ of the energy and $\frac{2 \cdot 137 + 4184}{1640 + 4184} = 77\%$ of the area. As we have eliminated the majority of the cost associated with the multiplication, the accumulation accounts for the majority of the remaining arithmetic costs of dot products. These costs can likely be reduced with methods such as chunked accumulation in lower precision [29, 35], combining multiple operations as done in tensor cores, or the Kulisch accumulation method used in [13]. We have not explored the use of these methods here but conjecture they are likely complementary to our approach, just as they have been for standard multiplication.

## C    Additional Architectures

The main focus of this work is on transformer networks. However we have found that piecewise affine matrix multiplications work well across a range of convolutional architectures. Table 5 shows the results of training five different non-transformer networks on CIFAR-10 using piecewise affine matrix multiplications with the approximate bwd pass, replacing all linear and convolutional layers.

Table 5: Top-1 Test Accuracy on CIFAR-10 for different networks when trained with standard matrix multiplications vs piecewise affine matrix multiplications (using approximate bwd). Each value is the average±std for three runs with different seeds. Brackets show the difference from the baseline.

| Network | Baseline | PA-Matmul |
|---|---|---|
| VGG-13 | $92.9 \pm 0.3\%$ | $92.9[+0.0] \pm 0.2\%$ |
| ResNet-20 | $92.1 \pm 0.3\%$ | $92.0[-0.1] \pm 0.2\%$ |
| ResNet-110 | $94.2 \pm 0.1\%$ | $94.1[-0.1] \pm 0.2\%$ |
| ResNeXt-20 4x16 | $93.7 \pm 0.2\%$ | $93.8[+0.1] \pm 0.2\%$ |
| ConvMixer-256/8 | $94.8 \pm 0.2\%$ | $94.8[-0.0] \pm 0.1\%$ |

In each case we roughly match the baseline (within the run-to-run variance), without any adjustments to the hyperparameters.

We selected the five models to demonstrate variety. Here are some of the ways in which they differ:

- VGG-13 [31]: Plain CNN without skip connection or normalization layers. Contains several fully connected layers at the end.
- ResNet-20 [11]: Model with skip connections and normalization.
- ResNet-110 [11]: Model of significant depth.
- ResNeXt-20 4x16 [37]: Uses bottleneck blocks and grouped convolutions.
- ConvMixer-256/8 [33]: Has a mixer style architecture and uses GeLU activations. The training setup also uses heavier data augmentation than for the other networks.

The exact details of the training setup used in each case can be found in our code.

## D   PAM with Narrower Mantissas

In this section we explore whether PAM can work with narrower mantissas which would decrease memory read costs and improve effective memory bandwidth. We simulate training with piecewise affine matrix multiplications (approximate bwd) using numerical formats with fewer mantissa bits than the 23 used in float32. We implement this by rounding the inputs and masking the extra mantissa bits, but do not change any other aspects of the training setup (i.e. no tuning or special low precision tricks). The internal accumulation is left unchanged similar to standard fp16-fp32 mixed precision. We focus on the training for two tasks, the IWSLT transformer from Section 3.2 and the VGG-13 CIFAR-10 network from Appendix C. The results can be seen in Table 6 below.

For both networks, we observe minimal to no discrepancy when training with 7 bits (equivalent to bfloat16). Mantissas as narrow as 4 bits work well, providing a comfortable margin for bfloat16 and offering promise for extensions into very narrow formats like 8-bit formats (although these might necessitate additional techniques to accommodate the narrow exponent). However, a 3-bit mantissa noticeably impairs the transformer's performance and may marginally impact the VGG training.

Table 6: The impact of the mantissa size of the PAM inputs on VGG-13 CIFAR-10 Test Accuracy and IWSLT14 BLEU Score for a small transformer. Each value is the average±std for three runs with different seeds.

| Matmul Type | CIFAR-10 Test Accuracy | IWSLT14 BLEU Score |
|---|---|---|
| float32 | $92.9 \pm 0.3\%$ | $34.4 \pm 0.1$ |
| PAM float32 | $92.9 \pm 0.2\%$ | $34.2 \pm 0.2$ |
| PAM bfloat | $92.9 \pm 0.2\%$ | $34.4 \pm 0.2$ |
| PAM 4 bit mantissa | $92.9 \pm 0.5\%$ | $34.2 \pm 0.1$ |
| PAM 3 bit mantissa | $92.8 \pm 0.2\%$ | $29.4 \pm 0.5$ |

## E   Runtime

Since PAM is not natively supported by the GPUs we use for our experiments, we require multiple instructions for each operation and can not make use of special hardware features such as the tensor

cores. We can also not use the highly optimized cuDNN and cuBLAS libraries at all, having to write the kernels required ourselves. Overall this can easily slow training down significantly compared to well optimized baselines that fully benefit from the GPU hardware including tensor cores. Here we want to emphasize that this is not a limitation of the method or our implementation but rather the lack of hardware support for the operation. We would incur a similar slowdown if we wanted to simulate training in a new custom floating point format, like some variant of float16, using only int and float32 operations. A significant effort went into enabling our experiments to run at the scale we report.

**IWSLT14 DE-EN**: The baseline runtime is around 1 hour on a V100 GPU. With PAM approximate matrix multiplications this number is around 4.5 hours. Replacing all operations results in a runtime of around 5.5 hours.

**DeiT i1k and CIFAR-10**: The CIFAR-10 baseline runs in about 4 hours on 4xV100 GPUs. The PAM approximate matrix multiplication version takes around 15 hours. The ImageNet version took 34 hours for the baseline vs 131 hours for PAM matrix multiplications on 6xA100 GPUs.

**Convolutional Networks on CIFAR-10**: The runtimes (baseline vs PAM matmuls) are roughly:

- VGG-13: 20 min vs 4.5 hours
- ResNet-20: 20 min vs 7 hours
- ResNet-110: 45 min vs 10 hours
- ResNeXt-20 4x16: 25 min vs 10 hours
- ConvMixer-256/8: 20 min vs 5 hours

We note that we did not focus on speeding up convolutions, electing to simply perform them as matrix multiplications using relatively inefficient folding operations.

# F    Limitations

The scale of typical deep learing problem makes it hard to investigate new low-level hardware algorithms in this field. The existing hardware accelerators that enable common large workloads are also highly specialized and do not necessarily perform well when performing slightly different operations. In this study the scale of the problems we can study is somewhat limited due to the high cost of training. We have tried to address this by writing custom kernels that allow us to run larger problems at a reduced cost.

Ideally, we could show direct hardware benefit from the piecewise-affine operations. However, we do not have the expertise to design an efficient hardware implementation that fully reaps the hardware advantages. Without significant investments, such an implementation would also be unlikely to allow training at the scale we demonstrate here. Instead, we have tried to estimate the potential hardware benefits and view this work as algorithmic evidence to inspire further work in this area.

