# OpenReview forum: "Multiplication-Free Transformer Training via Piecewise Affine Operations"
_NeurIPS.cc/2023/Conference — NeurIPS 2023 poster_

### Official Review · Reviewer_WjDg · 2023-07-02

**Soundness:** 3 good
**Presentation:** 2 fair
**Contribution:** 2 fair
**Rating:** 6
**Confidence:** 3

**Summary:**

This paper argues that multiplications are the main bottleneck in modern neural network training and inference, and proposes to reduce the cost by replacing them with a cheap piecewise affine approximation. This can eliminate all multiplications in the training and inference process as claimed.

**Strengths:**

* I think that this work has its value. It is a new type of multiplication-free network following addernets, shiftnets, or shiftaddnets. And it can be applied to both inference and training.

* The author has implemented a customized CUDA kernel for supporting their claims. And the final results are similar to the original networks.

**Weaknesses:**

* This paper claims to be the first work to reveal that a neural network has been trained entirely without standard multiplications. In my opinion, it overclaims as there are other multiplication-free networks with no multiplication involved during training or inference.

* The author only compares the accuracy but does not report the latency or efficiency metrics. That looks weird as the main motivation for adopting multiplication-free networks is to reduce the training or inference costs.

I also wonder about the speed comparison of the customized CUDA kernel and origin networks. Which will be faster (even for the FPGA case)? As in modern hardware, the computation is no longer the bottleneck as compared to data movements.

* How is the performance as compared to other multiplication-free/reduced networks in terms of accuracy and efficiency, e.g., addernet, deepshift, shiftaddnas, Ecoformer, and other binary neural networks?

**Questions:**

* Such a piece-wise linear approximation of multiplications can be seen as an approximation of a network. There are also other works that leverage piece-wise linear approximation to visualize and analyze the decision boundary or spline subdivisions. Are there any connections between these kinds of works? E.g., https://arxiv.org/abs/2302.12828, https://arxiv.org/abs/2101.02338

**Limitations:**

This paper gives an analysis of the limitation on the GPU speedups.

---

> ### Author Rebuttal · Authors · 2023-08-09
>
> We appreciate the time spent reviewing our submission as well as your feedback and suggestions! Below we try to address your concerns and questions.
>
> **Other multiplication-free works** We apologize if this is an overclaim, that was definitely not our intention. We are aware of other works e.g. AdderNet that can be multiplication free at inference which we mention in the paper. For training specifically, we are not aware of other schemes that are 100% multiplication free. We would be grateful if you can provide a reference and will include it in the paper and tone our claims down accordingly. Regarding the specific references you mention later in your review, although they are impressive works and able to remove most multiplications, they do include some leftover multiplications during training. They seem to use standard optimizers with the associated multiplications and do not try to address the multiplication in e.g. normalization when used. AdderNet relies on Batch Normalization including the learnable affine transformation. Deepshift can eliminate the multiplications involving the weights but if we understand correctly the computation of the gradient for a weight requires standard matrix multiplication. We believe ShiftAddNAS inherits this from the other papers. EcoFormer seems to address the attention matrix multiplication specifically but leaves the others unchanged. We will add these as references when we discuss related work.
>
> **Latency or efficiency metrics metrics** This is something we would have liked to include but the current lack of hardware support for these operations prevents us from giving a meaningful comparison. We currently simulate the arithmetic on GPUs using fp32 and int32 operations which will always be slower than the standard baseline. Note that this is not a limitation of the method itself, accurately simulating e.g. bfloat16 multiplication on hardware that only supports fp32 multiplications would also result in a similar slowdown. With proper hardware support we estimate a PAM operation to be around 5-10x cheaper than an fp16 multiplication in terms of energy and area (Appendix B). Other hardware elements required (that we do not focus on here) such as the accumulation and the memory will reduce the overall gains.
>
> **Comparison with other methods** While we can’t do an exhaustive comparison with all these methods we do compare to AdderAttention which is an extension of AdderNet for transformers (Section 3.2, Table 2). The AdderNet approach is very interesting and deviates from the traditional networks in a more drastic way. However, the PAM approach replaces more multiplications (all matrix multiplications), uses a cheaper replacement operation (integer additions instead of floating point) while resulting in a higher accuracy. We discuss some differences with the other works in the “other multiplication-free works” paragraph.
>
> **Memory Characteristics** Thank you for bringing this up. You are correct that our method only focuses on the cost of multiplication itself but other aspects of computation such as memory accesses also contribute to the overall cost. In the global response we discuss additional experiments that suggest that PAM is compatible with lower precision floating point formats that would result in memory savings as well as further computational savings. The large area savings from PAM could also be used for other purposes including additional memory.
>
> **Relation to other piece-wise linear works** This is a very interesting connection, thank you for pointing this out. Although their methods can already handle standard multiplications, it seems they are limited to piece-wise linear non-linearities such as leaky ReLUs. The piece-wise affine approximations like those we use for normalization layers and softmax might expand the applicability of their work to a broader range of architectures such as transformers, although we are not sure about the computational tractability in practice.

---

> > ### Comment · Reviewer_WjDg · 2023-08-19
> > **Response to the rebuttal**
> >
> > Thank the authors for the rebuttal. I will maintain my rating.

---

### Official Review · Reviewer_MZyL · 2023-07-04

**Soundness:** 2 fair
**Presentation:** 3 good
**Contribution:** 3 good
**Rating:** 5
**Confidence:** 3

**Summary:**

This paper proposes to replace all multiplications involved in a Transformer training process with bit additions of input floating-point representations. This is shown to be an approximation to the piecewise affine function that is again the approximation of common functions in Transformer training. Results show that this approximation will not cause obvious accuracy drop.

**Strengths:**

The idea of multiplication-free training is very interesting, and, if true, is significant. The bit-addition as an approximation is concise on both algorithm and hardware side.

**Weaknesses:**

My main concerns are detailed as follows:

1. There needs to be a better presentation in Section 2.2 on how piecewise affine multiplication (PAM) is reduced to bit-addition. At least the paragraph of line 106 is not obvious to the reviewer. For example, when EA=5, EB=5, SA=SB=MA=MB=0, the two floating-point numbers (now actually integers) A * B = 32 * 32 = 1024, but EA+EB+MA+MB+SA+SB = 10, which clearly doesn't equal and the results have a large gap.

2. In terms of performance (latency), is the bit-addition of floating-point representations (population count) better than multiplication? The Wallace tree implementation of multiplication has a time complexity of O(log(b)), seems like the same as the bit-addition.

3. Is the proposed method compatible with low-precision integer/floating-point formats? What would be the comparison between a quantized matmul and PAM?

4. The setup of results in Table 2 is confusing. It reports the training accuracy with only the matmul replaced with PAM, which is the same as the prior work Mogami (2020). What are the results with all layers replaced with PAM, especially with all optimizer ops replaced?

5. For Table 3 machine translation results, what are the model architecture details, e.g., number of parameters, layers, heads, etc.? What is the activation function used in the FFN (GELU or ReLU)? Is it replaced with PAM as well? It is also better to report loss values besides BLEU since BLEU is typically noisy.

**Questions:**

Questions are listed together with the main concerns in the previous section.

**Limitations:**

The reviewer is not aware of any social impact of this paper.

---

> ### Author Rebuttal · Authors · 2023-08-09
>
> Thank you for your time and effort in reviewing our paper! Below we try to address your concerns and feedback.
>
> **(1) PAM as bit addition** We apologize that this can be misunderstood and propose rephrasing this paragraph as: We observe that Aˆ· B is roughly achieved by adding the signs (Equation 6), exponents (Equation 7), and mantissas (Equation 8) of A and B. If we add the floating point representations of A and B given in Equation 3 together as integers, the extra term of 1{M_A + M_B>1} in Equations 7 and 8 corresponds to an overflow of the resulting $\bar{M}$ into $\bar{E}$. Piecewise affine multiplication can therefore be performed by an int32 addition of the floating point representations, barring some technical details we discuss next.
>
> We hope this clarifies that in your example the addition would be carried out as PAM([0, 5, 0], [0, 5, 0]) = [0+0, 5+5, 0+0] = [0, 10, 0] = 1024 which gives the exact result (this is generally the case when one operand is an exact power of two).
>
> **(2) Latency** This is not our area of expertise so we apologize if we misunderstood your point or if there are errors in the following answer. If we understand correctly, a Wallace tree implementation concerns the implementation of an integer multiplication through a tree-like reduction of the partial products. The depth of the tree is O(log2(b)) where b is the bit width of the integers. Each addition in the tree operates on integer inputs with a width of at least b. The PAM operation (for floats) can be expressed as a single full width int addition (similar to one level of the Wallace tree) followed by a fix of the exponent (int8 addition) and handling for underflow / overflow of the exponent. In Appendix B we approximate the total cost as being around 2 full-width integer additions. If this holds, the latency could perhaps be modeled as around 2 levels of the Wallace tree resulting in a lower latency. Even if this turns out not to be the case, the resulting area savings would still be beneficial by freeing up area for other uses.
>
> **(3) Lower precision floats** Yes, we believe the method should work with lower width floats such as bfloat16. We have added a global response to discuss additional experiments that support this. We believe for 16 bit formats PAM should be roughly 5-10x cheaper than standard multiplication.
>
> **(4) Table 2** In Section 3.2 and Table 2 we only replace the matrix multiplication. This allows us to compare directly to AdderAddention, an alternative approach for replacing multiplications in transformer training, showing better performance while replacing more multiplications. In Section 3.3, 3.4 and Table 3 we extend this, studying the impact of replacing all operations individually and cumulatively. This is done on a computationally cheaper task allowing us to run different combinations and obtain error bars.
>
> **(5) Table 3** In the manuscript we describe the setup used in this experiment in Lines 204-213. The network is based on the original transformer architecture and has 6 encoder and 6 decoder layers with ReLU activations. Thank you for the suggestion of additionally reporting the loss. It seems like a good idea in general but in this we think it could potentially be misleading since we use a different loss function for some of the table entries (i.e. the piecewise affine approximation).
>
> We will incorporate your feedback, clarifying the points and adding the results for lower bit widths. If you feel we have sufficiently addressed some of your concerns, we would greatly appreciate it if you would consider raising your review score.

---

> > ### Comment · Reviewer_MZyL · 2023-08-19
> > **Response to rebuttal**
> >
> > Thanks for addressing the comments.

---

### Official Review · Reviewer_aJYh · 2023-07-05

**Soundness:** 3 good
**Presentation:** 3 good
**Contribution:** 2 fair
**Rating:** 5
**Confidence:** 3

**Summary:**

The paper introduces a novel approach that replaces all multiplications in Transformer training with a cost-effective piecewise affine approximation achieved by adding the bit representation of floating-point numbers. This method allows for a full multiplication-free training of Transformer models, covering linear and nonlinear operations in the forward, backward, and optimization phases. The authors demonstrate that this approximation leads to minimal accuracy degradation across different training scenarios and provide estimates of the potential savings in terms of area and power requirements.

**Strengths:**

* The paper introduces a new "multiplication-free training" scheme that can be applicable to Transformer models and other deep learning architectures.

* Despite its simplicity, the approximation method shows only small gaps and minimal accuracy degradation compared to real values.

* The paper provides a unique way of reducing training costs.

**Weaknesses:**

While the underlying motivation of replacing all the multiplications with simpler operations throughout the entire training procedure, and the proposed method is appealing, it entails some concerns:

1. It looks like the general concept of piecewise affine approximation is not new and was introduced in Mogami. The authors stress (in L317-324) that the main difference from Mogami is the extension of the piecewise affine approximation to include forward, backward, and optimization, enabling a multiplication-free training procedure. However, the benefits of a multiplication-free network are not so clear for multiple reasons:

  * (1-1) The benefit of applying multiplication-free approximation to non-linear operations is unclear. Non-linear operations are generally compute-bound and constitute only a small portion of the overall inference runtime. Since the proposed method can only reduce the compute cost (not the memory cost), the gain from carrying out those computations without multiplications can be minimal.
   * (1-2) The suggested benefit of allowing the model to be deployed on hardware without multiplication support (L324) is unclear. The author should provide some real examples, or otherwise, it appears hypothetical. Furthermore, even if we were to design new multiplication-free hardware from scratch, it remains uncertain whether the advantages (in terms of saving area, power, and latency) would be substantial enough to outweigh the noticeable performance degradation from matrix-free optimization and loss computation, as indicated in Table 3.
That being said, the author should provide a more thorough comparison between computing linear operations without multiplication (as in Mogami - baseline) and computing the entire training process without multiplication to differentiate their work from prior research.
Otherwise, it seems more favorable to run Mogami's partial PAM scheme on hybrid hardware with small multiplication units (just so that they can support a few non-heavy linear operations), which would offer a better trade-off between runtime costs and accuracy, as compared to the proposed methodology.
   * (1-3) Furthermore, since the proposed methodology is targeting training  (where the device must support multiple models) rather than application-specific inference, designing new hardware would require flexibility. The proposed solution relies on multiplication-free hardware, which would restrict the broader applicability of training various/new model architectures (e.g. with new nonlinearity).



2. The training cost comprises both compute operations and memory operations. Although the authors provide a reasonable estimate of how the proposed method reduces compute cost, it does not mitigate memory cost as all values are still stored in 32-bit precision.
For instance, in Figure 7 of [2], which serves as a reference for Table 4 in the paper, it is demonstrated that memory operations consume approximately two orders of magnitude more energy than arithmetic operations even when loading from SRAM.
Given that the memory operations can constitute a large portion of the overall runtime cost and the proposed method is not so effective at reducing the cost of memory load/store, the estimate of the overall energy saving provided in the paper remains unclear.
i.e., If a significant portion of the end-to-end energy consumption comes from memory operations, the savings achieved through computation reduction might be minimal.

3. Several methodologies have been proposed for efficient training using reduced-precision approaches (such as bfloat16, which has already been settled down as the norm [3], or integer-only training [4] even though it is not for Transformer training), and this should also be considered as the baseline to compare against.
These methods not only decrease compute costs but also reduce memory costs.
In terms of end-to-end energy and latency reduction, would the proposed method offer greater benefits compared to these existing methods?
Considering the additional requirement for kernel/hardware design of the proposed method (versus reduced-precision training), the author should provide enough evidence that proves the overall gain to be considerably better than those methods to convey its practical value.

[1] Full Stack Optimization of Transformer Inference: a Survey, https://arxiv.org/pdf/2302.14017.pdf

[2] A Survey of Quantization Methods for Efficient Neural Network Inference, https://arxiv.org/pdf/2103.13630.pdf

[3] A Study of BFLOAT16 for Deep Learning Training, https://arxiv.org/pdf/1905.12322.pdf

[4] NITI: Training Integer Neural Networks Using Integer-only Arithmetic, https://arxiv.org/pdf/2009.13108.pdf

**Questions:**

- Do authors have any insights on why the exact backward functions yield more training instability and worse performance than the approximated ones?

**Limitations:**

Please see the weakness section

---

> ### Author Rebuttal · Authors · 2023-08-09
>
> Many thanks for your time and the feedback you've given. Below we try to address the main concerns you list one by one.
>
> **Fully multiplication free (1)** You are correct that from a hardware perspective focusing on the matrix multiplications (on both the forward and backwards passes) may be sufficient to reap most of the benefits for certain network and hardware architectures. In this work we show an approach like this can work across a variety of architectures including transformers (which has not been shown previously) before extending it to fully multiplication-free training. We also release code to enable further exploration of this area.
>
> Being fully multiplication free may in some ways be of more academic interest for now (for current hardware and architectures). We still believe it is a valid question to ask given the complete reliance of current hardware and training schemes on multiplications. We show that this does not need to be the case and training entirely without multiplications is in fact viable. Future networks and training schemes could enhance its practical relevance. For example, architectures with a more limited connectivity which skew the ratio of matrix computation to other operations might be of interest on new hardware. The benefit of depth-wise separable and grouped convolutions over their dense variants could indicate that this is the case. The loss function is the only significant contributor to the modest performance decrease from the fully multiplication free scheme and affects a network with standard multiplications the same way. Deviating further away from the standard loss function might mitigate this, e.g. using base 2 logarithms and exponents which we can approximate much better using the piecewise linear framework (we discuss this as a future direction in L345). The fully piece-wise affine approximations might also be of interest in other non-hardware related areas such as the network analysis brought up by Reviewer WjDg.
>
> It is also correct that training hardware might need to support a variety of possible models. Hardware such as FPGA could potentially perform fully multiplication free training while still supporting a variety of architectures. The fully multiplication free approach also extends to inference applications where specialized accelerators may be more feasible.
>
> **Memory Costs and Reduced Bitwidths (2) and (3)** Thank you for pointing this out. We have added an experiment (see global response) that suggests that PAM can be used with narrower formats such as bfloat16 and even fewer mantissa bits. This should give memory access and bandwidth savings similar to other approaches as well as further reduce the computational costs. The area savings from the cheaper multiplication could also be used for other purposes such as memory.
>
> **Exact vs approximate bwd** The approximate backward functions seem to better approximate the gradients of true multiplication in some ways. The exact derivatives give unbiased gradient estimates (of true multiplication) but can deviate more at a given location. The approximate derivative is potentially biased but typically closer to the true multiplication derivative. Figure 3 (Appendix A) plots the two types of derivatives for a visual comparison. Since PAM approximates multiplication over the long term, the gradients of true multiplication could better describe how the loss surface changes over longer distances. These gradients, and by extension the approximate gradient, could therefore serve as a “smoothed” or denoised gradient that aids optimization.
>
> **General remark** In this manuscript, we have explored the question of whether neural networks can be trained in a fully multiplication-free fashion. We believe this is an interesting academic question in itself and could also hold practical relevance, whether by focusing solely on matrix multiplications (as demonstrated for various new architectures) or by fully eliminating all multiplications. The space savings achieved from the more economical multiplication replacements could be allocated to other purposes, including bandwidth and memory improvements. While we don't address every issue involved in the bigger picture, we believe our results present a viable path for future hardware improvements and are of interest to the community. We will include the results for the narrower floating-point formats in an appendix. If you feel that we have addressed some of your concerns, we would greatly appreciate it if you would consider slightly increasing your review score.

---

> > ### Comment · Reviewer_aJYh · 2023-08-19
> >
> > I appreciate the clarifications made in the rebuttal.
> >
> > **Fully multiplication free**: I still find the multiplication-free scheme somewhat theoretical and academic for now, and adding concrete examples of hardware architectures would have strengthened the idea even further. Nevertheless, I agree with the author's claim about the potential advantages and future prospects of the suggested scheme.
> >
> >
> > **Memory Costs and Reduced Bitwidths:** We appreciate the authors for the added experiments/results, which will strengthen the submission.
> >
> >
> > Overall, some of my concerns about the paper have been addressed, so I have raised my score to 5.

---

### Official Review · Reviewer_DF7v · 2023-07-06

**Soundness:** 3 good
**Presentation:** 3 good
**Contribution:** 3 good
**Rating:** 7
**Confidence:** 3

**Summary:**

This paper presents a method for training deep networks completely without multiplication, via approximating multiplication using piecewise affine operations. The authors show that their method can be used to train modern deep networks, including Transformers.

**Strengths:**

The paper is **extremely** interesting. The ideas are great, and the findings are quite surprising. The results are preliminary but seem quite promising, and the idea is worth exploring in more depth. A future where all neural networks are trained without multiplication sounds very interesting and exciting!

**Weaknesses:**

Modern deep networks are supported by extensive hardware support, such as tensor cores for matrix multiplication. This means that the algorithms proposed in this paper, although theoretically more efficient, are not more efficient in practice. The paper would be stronger if it were more upfront with these limitations, and measured wallclock of the algorithms on modern hardware. The paper would be stronger with this comparison upfront, especially since "Hardware-Efficient" is in the title (an alternate framing, such as "Multiplication-Free Transformer Training via Piecewise Affine Operations" would not suffer from this same weakness).

**Questions:**

1. What is the wallclock runtime of this method? How would it change with the equivalent of tensor core support for this approximation?

**Limitations:**

Limited discussion of wallclock characteristics on modern hardware.

---

> ### Author Rebuttal · Authors · 2023-08-09
>
> Thank you for the time spent reviewing our submission and as well as your feedback and suggestions!
>
> We agree that it would likely have been better to de-emphasize the hardware efficiency and focus more on the multiplication-free aspect. We are unfortunately unable to change the title here, but will give this a serious consideration in the case it is not accepted here.
>
> We try to discuss the theoretical hardware benefits and current runtime in Appendices B, D and E. Due to the lack of hardware support for PAM operations on GPUs we simulate the PAM arithmetic using multiple INT32 and FP32 instructions for each multiplication. This results in runtimes that are several times slower than an FP32 baseline on the GPUs used. We still think this is a relatively good runtime for a detailed simulation of arithmetic that is not supported in the hardware and spent considerable time on the kernels to enable this. With proper hardware support and a similar tensor core implementation we estimate that the multiplication itself would be on the order of 5-10x cheaper in terms of hardware area and energy cost compared to FP16 (Appendix B). This would allow packing more processors etc on a given chip which should result in increased speeds / wallclock runtime. Other hardware elements required (that we do not focus on here) such as the accumulation and the memory will reduce this number but we believe these can be addressed using orthogonal methods such as narrower floating point formats (global response) and the accumulation approaches discussed in Appendix B.

---

> > ### Comment · Reviewer_DF7v · 2023-08-11
> >
> > Thank you for your response! I recommend moving part of this submission to the main paper for the next version.

---

### Author Rebuttal · Authors · 2023-08-09

# Global Response
We are grateful to the reviewers for their efforts, insightful comments and constructive suggestions. We respond to all reviews individually. In this response we discuss the compatibility of PAM with lower precision formats, a question raised by several reviewers. In the manuscript we mention that we believe PAM should work with narrower mantissas but did not perform experiments to validate this. Narrower mantissas would yield further computational savings as well as memory savings (including storage, read costs and bandwidth).

To address this concern we have performed an additional experiment where we simulate training with piecewise affine matrix multiplications (approximate bwd) using numerical formats with fewer mantissa bits than float32. We achieve this by rounding the inputs and masking the extra mantissa bits but do not change any other aspects of the training setup (i.e. no tuning or special low precision tricks). The internal accumulation is left unchanged similar to standard fp16-fp32 mixed precision. We focus on the training for two tasks, the IWSLT transformer from Section 3.2 and the VGG-13 CIFAR-10 network from Appendix C. The results can be seen in the table below (average±std for three runs).

For both networks, we observe minimal to no discrepancy when training with 7 bits (equivalent to bfloat16). Mantissas as narrow as 4 bits work well, providing a comfortable margin for bfloat16 and offering promise for extensions into very narrow formats like 8-bit formats (although these might necessitate additional techniques to accommodate the narrow exponent). However, a 3-bit mantissa noticeably impairs the transformer's performance and may marginally impact VGG training.

| Matmul Type      | VGG-13 Test Accuracy | IWSLT14 BLEU Score |
| ----------- | ----------- | ----------- |
| float32 | 92.9±0.3% | 34.4±0.1 |
| PAM float32 | 92.9±0.2% | 34.2±0.2 |
| PAM bloat16 | 92.9±0.2% | 34.4±0.2 |
| PAM 4 bit mantissa | 92.9±0.5% | 34.2±0.1 |
| PAM 3 bit mantissa | 92.8±0.2% | 29.4±0.5 |

We will add these results to a new appendix section.

---

### Decision · Program_Chairs · 2023-09-21

**Decision:**

Accept (poster)

**Comment:**

The reviewers all recommend acceptance. Having looked at the paper, reviews, and rebuttal, we are accepting this paper. The authors are encouraged to improve the final paper version by following reviewer recommendations.